# The *TP53* Codon 72 Arginine Polymorphism Is Found with Increased *TP53* Somatic Mutations in HPV(−) and in an Increased Percentage among HPV(+) Norwegian HNSCC Patients

**DOI:** 10.3390/biomedicines11071838

**Published:** 2023-06-26

**Authors:** Svein Erik Moe, Fredrik A. Erland, Siren Fromreide, Stein Lybak, Marianne Brydoy, Harsh N. Dongre, Sophia M. Dhayalan, Daniela-Elena Costea, Olav K. Vintermyr, Hans Jørgen Aarstad

**Affiliations:** 1Department of Otolaryngology/Head and Neck Surgery, Haukeland University Hospital (HUS), N-5020 Bergen, Norway; svemoe@helse-bergen.no (S.E.M.); fredrik.atteraas.erland@helse-bergen.no (F.A.E.); slyb@helse-bergen.no (S.L.); 2Department of Clinical Medicine, University of Bergen, N-5020 Bergen, Norway; siren.fromreide@uib.no (S.F.); harsh.dongre@uib.no (H.N.D.); daniela.costea@uib.no (D.-E.C.); 3Department of Oncology, Haukeland University Hospital (HUS), N-5020 Bergen, Norway; marianne.brydoy@helse-bergen.no; 4Department of Pathology, Haukeland University Hospital (HUS), N-5020 Bergen, Norway; sophia.manueldas.dhayalan@helse.bergen.no (S.M.D.); olav.vintermyr@helse-bergen.no (O.K.V.)

**Keywords:** head and neck cancer, *TP53*, somatic mutation, singe-nucleotide polymorphism, HPV

## Abstract

Background: Somatic *TP53* mutations are frequent in head and neck squamous cell carcinoma (HNSCC) and are important pathogenic factors. Objective: To study *TP53* mutations relative to the presence of human papillomavirus (HPV) in tumors in HNSCC patients. Methods: Using a custom-made next-generation sequencing (NGS) panel on formalin-fixed, paraffin-embedded tumor tissue, we analyzed somatic *TP53* mutations and the *TP53* single-nucleotide polymorphism (SNP) codon 72 (P72R; rs1042522) (proline → arginine) from 104 patients with HNSCC. Results: Only 2 of 44 patients with HPV-positive (HPV(+)) HNSCC had a *TP53* somatic mutation, as opposed to 42/60 HPV-negative (HPV(−)) HNSCC patients (*p* < 0.001). Forty-five different *TP53* somatic mutations were detected. Furthermore, in HPV(−) patients, we determined an 80% prevalence of somatic *TP53* mutations in the *TP53* R72 polymorphism cohort versus 40% in the *TP53* P72 cohort (*p* = 0.001). A higher percentage of patients with oral cavity SCC had *TP53* mutations than HPV(−) oropharyngeal (OP) SCC patients (*p* = 0.012). Furthermore, 39/44 HPV(+) tumor patients harbored the *TP53* R72 polymorphism in contrast to 42/60 patients in the HPV(−) group (*p* = 0.024). Conclusions: Our observations show that *TP53* R72 polymorphism is associated with a tumor being HPV(+). We also report a higher percentage of somatic *TP53* mutations with R72 than P72 in HPV(−) HNSCC patients.

## 1. Introduction

More than 900,000 new cases of head and neck cancer (HNC) (lip, oral cavity, pharynx, larynx, or salivary glands) were reported worldwide in 2020, accounting for approximately 5% of the total incidence of major cancers. Worldwide, approximately 50% of newly diagnosed HNC patients survive 5 years following diagnosis [1]. In 2020, the incidence of HNC in Norway accounted for approximately 2.5% (*n* ≈ 800) of total cancer incidence, with curative treatment reported for approximately two-thirds of diagnosed patients [2].

The consumption of alcohol and tobacco, especially in combination, is a well-known risk factor for the development of head and neck squamous cell carcinoma (HNSCC) [3]. Another risk factor for HNSCC is poor oral health [4]. In recent decades, investigators have firmly concluded that the human papillomavirus (HPV) virus is a causal agent in HNSCC, especially in oropharyngeal (OP) SCC. The incidence of OPSCC is increasing [5]. Furthermore, cancers with HPV-positive (+) tumors exhibit different biology than HPV-negative (−) tumors, with HPV(+) patients having a better prognosis than HPV(−) OPSCC patients [6], along with a different mutational profile [7].

HPV is a small, double-stranded, circular DNA virus that may infect the epithelial cells in the head and neck region. The most well-known oncogenes from high-risk subtypes of HPV are undoubtedly its E6 and E7 oncogenes [8]. E6 binds strongly and avidly to *TP53*, forming a complex with a ubiquitylation protein, E6-AP, which downregulates its tumor-suppressor functions [9]. The E7 protein, on the other hand, binds avidly to retinoblastoma (RB)-associated proteins 1 and 2, which drives the proteasomal destruction of RB and the release of the E2F family of transcription factors [10]. These E2F proteins push the cell cycle beyond the G1-S checkpoint into the S phase [11]. The E7 dysregulation of RB function leads to positive feedback upregulation of p16^INK4A^ in HPV(+) tumors [12]. On the other hand, HPV(−) tumors have a very different cause, making their detailed investigation of paramount interest.

The genetic landscape of HNC includes various somatic mutations and chromosomal aberrations [13], as well as copy number variations and epigenetic changes [14]. The molecular mechanisms underlying carcinogenesis in classical HNSCC have been unraveled to some extent [15]. Carcinogenesis is considered a multistep process from dysplasia to cancer [16]. Reports published on the genomic analysis of HPV(−) HNSCC demonstrated findings of mutations such as *TP53*, CDKN2A, MLL2/3, NOTCH1, PIK3CA, NSD1, FBXW7, DDR2, and *CUL3* [13,15,17,18]. The reported mutational profile varies to some extent, but *TP53* is the most common mutated gene in HNSCC [18,19].

The p53 tumor-suppressor system participates in a variety of essential cell functions, such as cell cycle arrest, cellular senescence, DNA repair, apoptosis, autophagy, cell metabolism, immune system regulation, generation of reactive oxygen species, mitochondrial function, and global regulation of gene expression [19,20]. Furthermore, the mutation of the *TP53* tumor-suppressor gene is the most common genetic alteration in human cancer [21]. Many different *TP53* allelic variants have been reported in human tumors [21]. Cells with *TP53* mutations may lose the ability to execute wild-type p53 functions to varying degrees [22] and act as dominant negative inhibitors of wild-type p53 tumor-suppressive functions [23].

Several well-characterized single-nucleotide polymorphisms (SNPs) have also been reported in the *TP53* gene. The common *TP53* codon 72 SNP (P72R; rs1042522) is one of the most thoroughly studied [24,25]. The *TP53* codon 72 SNP involves a change from cytosine to guanine in the coding sequence, resulting in a change from proline (P72) to arginine (R72) in the protein [26]. The *TP53* R72 SNP has been reported to modulate the ability to induce apoptosis [27]. Furthermore, the E6 oncoprotein of high-risk HPV virus has been hypothesized to improve the degradation of *TP53*, which harbors arginine instead of proline [28], with the potential to modulate the risk of OPSCC in HPV-infected patients.

Yeast transcription assays indicate more *TP53* mutations are associated with the codon 72 arginine than proline relative to their prevalence in the germline [29]. Various studies have also shown that this *TP53* SNP may affect apoptosis and cell cycle progression [27,30] by, e.g., enhancing the metastatic potential of mutant *TP53* in cell lines with the *TP53* R72 (arginine) SNP compared to mutated cell lines with the ancestral *TP53* P72 (proline) SNP [31]. In support of this phenomenon, cell lines harboring some specific somatic *TP53* mutations have been shown to exhibit more aggressive growth in association with the *TP53* R72 SNP than the *TP53* P72 SNP [32]. Furthermore, an increased prevalence of the *TP53* R72 SNP has been observed in humans living in higher latitudes and colder climates [33].

The aim of this study was to identify the rate and repertoire of *TP53* somatic mutations and the prevalence of the *TP53* R72 (arginine) SNP, as well as the relationship between these parameters in a cohort of both HPV(+) and HPV(−) HNSCC patients. Such results may explain why some acquire HNSCC.

## 2. Materials and Methods

### 2.1. Patient Population and Study Design

Since 1992, all patients with head and neck cancer (HNC) at the Department of Otolaryngology, Head and Neck Surgery, Haukeland University Hospital (HUH), Bergen, Norway, have been registered in a hospital-based HNC registry. HUH treats all cases of HNC in the Western Health Care Region of Norway. This region includes 1.1 million inhabitants.

All HNSCC patients were subjected to a standardized diagnostic workup, including clinical examination; CT/MRI scans of the primary tumor site, neck, thorax and liver; and ultrasonographic examination of the neck with fine-needle aspiration cytology if indicated. If possible, a diagnostic endoscopic examination was performed under general anesthesia. From this cohort, we extracted 104 patients diagnosed during the period from 2003 to 2016 for further analysis. The clinical data for each patient were obtained from a retrospective chart review. The tumor site was classified according to the International Classification of Diseases (ICD), 10th edition. The TNM stage was classified according to the International Union against Cancer (IUCC), 7th edition.

### 2.2. Protocols for HPV Detection

For the detection of HPV DNA, standard Gp5+/Gp6+ primers were used, as previously described [6,34].

### 2.3. NGS Panel Preparation and Analysis

DNA isolation, HPV detection, and our custom-made next-generation sequencing (NGS) panel were previously described [17]. Briefly, an experienced pathologist selected representative tumor samples. The tumors represented primary HNC diagnostic or surgical samples collected in the diagnostic workup. DNA was isolated from formalin-fixed, paraffin-embedded (FFPE) blocks. DNA was extracted using a commercially available DNA extraction kit according to the manufacturer’s protocol (E.Z.N.A tissue DNA kit, Omega BioTek, Norcross, GA, USA), and the DNA concentration was quantified (Qubit dsDNA BR assay, Thermo Fisher Scientific, Waltham, MA, USA). Then, 50 ng of DNA was used to prepare amplicon libraries using an AmpliSeq Library PLUS kit (Illumina, San Diego, CA, USA). Thereafter, the DNA was purified with AMPure XP beads (Beckman Coluter, High Wycombe, UK). To reduce the fixation impact, the two last steps were repeated before final quantification (Qubit dsDNA HS Kit, Thermo Fisher Scientific, Waltham, MA, USA). In the end, 1.3 pM of the library was loaded for paired-end sequencing using the Illumina MiniSeq platform, and further data management was performed in BaseSpace Sequence Hub (Illumina, San Diego, CA, USA).

Bioinformatics analysis was performed using the DNA amplicon v2.2.1 workflow in BaseSpace (Illumina, San Diego, CA, USA). The full coding sequence of *TP53* was aligned to the hg19/GRCh37 reference genome using the Burrows–Wheeler aligner. Variants were called using the somatic variant caller and annotated by RefSeq using the National Centre for Biotechnology Information (NCBI) database. A total amplicon depth (coverage) of more than 500 reads and a variant allele (VAF) of 5% were set as strict thresholds. Only coding sequences were examined. The recorded variants were evaluated in the dbSNP (NCBI) and Catalogue of Somatic Mutations in Cancer (Cosmic) databases.

### 2.4. Statistical Analyses

Statistical analysis was performed using the SPSS statistical program (IBM Corp., IBM SPSS Statistic for Windows, Version 26.0, Armok, NY, USA). Student’s *t*-test and the Mann–Whitney test were used to compare groups. The *p*-value for the two-sided test is reported.

## 3. Results

### 3.1. Clinical Variables

The present cohort (*n* = 104) represents a subgroup of the entire group of patients diagnosed with HNSCC in western Norway during the period from 2003 to 2016. The primary tumor sites were the oropharynx (OP) and the oral cavity (OC) (Table 1). Table 1 shows the age, gender, and TNM stage of patients in the studied cohort sorted by tumor HPV status. Among the 104 included patients, 60 had HPV(−) tumors, and 44 had tumors that were HPV(+). Almost all HPV(+) tumors were positive for HPV type 16, except for four tumors, including two tumors that were positive for HPV18, one that was positive for HPV33, and one that was positive for HPV58.

### 3.2. TP53 Somatic Mutations Detected in the Study Cohort

The mean age among the HPV(−) tumor patients was 63.1 ± 11.3 years versus 59.4 ± 10.1 years among the HPV(+) tumor patients. Regarding the TNM stage, only the N stage differed between the HPV(−) and HPV(+) tumor groups (Table 1). Smoking history did not statistically significantly differ between HPV(+) and HPV(−) patients (Table 1).

A total of 45 different *TP53* mutations were registered (Appendix A). Of the detected single-base mutations, 36 were pathogenic, and 3 were neutral according to Functional Analysis through Hidden Markov Model (FATHMM) software version 2.3 [35] (Appendix A). In addition, six were insertions/deletions. A total of 44 of 104 patients had somatic *TP53* mutations (Table 2, upper frame). Furthermore, 34 patients had 1 *TP53* somatic mutation, 9 patients had 2, and 1 patient had 3 (Table 3, upper frame).

Of patients with tumor somatic *TP53* mutations, 42 patients were HPV(−), and two were HPV(+) (Table 2), which is a highly significant difference (*z* = −6.64; *p* < 0.001). Among the HPV(−) patients, 33 had one somatic *TP53* mutation, 8 patients had two *TP53* mutations, and one patient had three *TP53* mutations (Table 3). Of the 44 patients with HPV(+) tumors, one had one *TP53* mutation, and one had two (Table 3), constituting a highly significant difference between the HPV(+) and HPV(−) tumors (*z* = −6.38; *p* < 0.001).

### 3.3. Detection of TP53 P72R in HPV(+) and HPV(−) HNSCC Patients

The *TP53* somatic mutations were distributed differently among the HPV(−) patients when comparing OP (15/27 cases) versus OC (24/29 cases) origin (*z* = −2.19; *p* = 0.028) (Table 4). Moreover, for HPV(−) patients, the presence of a *TP53* mutation was associated with the *TP53* R72 polymorphism (Table 2) (*z* = −3.41; *p* < 0.001). For OPSCC HPV(−) (*z* = 2.01; *p* = 0.044) and OC (*z* = 2.27; *p* = 0.023) patients, there was also a preference for R72 regarding the number of *TP53* mutations (Table 3).

*TP53* R72 polymorphism was associated with an increased presence of somatic *TP53* mutations in HPV(−) patients. Among the 18 HPV(−) *TP53* mutant-negative patients, only 7 harbored the *TP53* R72 SNP, whereas among the 42 HPV(−) *TP53* somatic-mutated patients, 35 patients harbored this SNP (Table 2) (*z* = −3.41; *p* < 0.001). In addition, when the number of TP53 somatic mutations was included in the analyses, the *TP53* R72 SNP was also associated with *TP53* mutations (Table 3) (*z* = −2.85; *p* = 0.004).

In HPV(+) HNSCC patients, there was a preponderance of patients with the *TP53* arginine codon 72 polymorphism. A total of 39 of 44 patients had this SNP in the HPV(+) group, whereas 42 of 60 carried this SNP in the HPV(−) group of patients (z = −2.25; *p* = 0.024) (Table 5). Among the OPSCC patients only, the results were similar (*z* = −2.24; *p* = 0.025) (Table 5).

Among the five HPV(+) SCC patients with the *TP53* P72 SNP (proline), one patient had a somatic *TP53* mutation (Table 2). In two of these patients, pathogenic mutations were observed in the *TRAF3* gene and in the *FGFR3* gene, respectively (Table 6). In the other three patients, no further mutations were detected with the employed NGS panel [17] (Table 6).

## 4. Discussion

*TP53* has a complex interplay with several cellular pathways, including key roles in cell function [20], maintaining genetic stability [25], and carcinogenesis when mutated [36]. Somatic mutations in the *TP53* gene are a common cancer feature [37,38]. Most mutations in the *TP53* gene are single-base missense mutations altering the encoding amino acid [38]. Several *TP53* mutations tend to be clustered (so-called hotspot mutation), as reported in HNSCC [22]. The *TP53* gene codes for one protein with many possible *TP53* mutations [21]. Our NGS panel had full coverage of the *TP53* coding sequence, including all common *TP53* hotspot somatic mutations [17]. In addition, the tumor microenvironment, both within the tumor cells and the cellular tumor landscape, is important [39,40,41]. Regarding prognosis and treatment responses, HNSCC tumors have especially been shown to interact with the general immune system [42,43].

In the present study, many different and well-characterized *TP53* somatic mutations were encountered in the HPV(−) cohort of HNSCC patients. Some patients in this group also harbored multiple somatic *TP53* mutations. In the HPV(+) group of patients, only one pathogenic *TP53* missense mutation was encountered, which is likely related to the well-established effect of HPV viral oncoprotein E6, enabling the capture of the host p53 tumor suppressor protein [44]. Furthermore, tobacco smoking has been associated with increased levels of *TP53* somatic mutations in HNSCC [45], but there was no significant difference with respect to smoking habits in the HPV(+) versus HPV(−) cohort of patients in our study. This could support that infection with HPV virus protects against *TP53* mutations.

In the reported study, mutations in the *TP53* gene were analyzed, with an additional focus on the *TP53* codon 72 SNP, resulting in an arginine-reading codon from a proline-reading codon (rs1042522). Ancestral *TP53* P72 occurs more frequently in equatorial human populations, whereas the TP53 R72 polymorphism occurs more commonly at higher latitudes, including in the Norwegian population [24]. In this respect, we observed a 73.5% prevalence of *TP53* R72 SNPs in HPV(−) HNSCC patients, which seems to be at the high end of various European population estimates [46]. Furthermore, *TP53* R72 SNPs were detected in most HPV(+) HNSCC patients, with the exception of five patients, giving an 89% prevalence among the HPV(+) patients, which is a significantly higher preponderance of this particular SNP than among the HPV(−) patients.

A limitation of our study is that analyses were performed only in tumor cells, as the *TP53* R72 SNP in heterozygous individuals may be selected for tumor cells. In principle, the reported observations may be influenced by a loss of heterozygosity [32,47]. We have, however, not differentiated *TP53* allele status in our analyses.

A major observation from our study is that in HNSCC HPV(−) patients, somatic *TP53* mutations were associated with *TP53* R72 SNPs. *TP53* R72 SNPs have previously been associated with HNSCC in a Brazilian study [48]. An association of *TP53* R72 SNPs with oral cancer has also been reported [49]. However, in a study by Jiang et al., no such association was found [50]. Previous studies have also reported an association of *TP53* R72 SNPs with laryngeal cancer [51]. In yet another recent study, Escalante et al. suggested that *TP53* R72 SNPs may be a risk factor for the pathogenesis of laryngeal cancer [52]. In addition, in the upper gastrointestinal tract, an increased risk of esophageal SCC has been associated with *TP53* R72 SNPs [53], as also suggested regarding lung and breast cancer in a South Asian population [54]. In other HPV-associated cancers, such as uterine cervical cancer, *TP53* R72 SNPs have been suggested to be more common in HPV(+) than HPV(−) cancers [55]. In the above-mentioned study, an association with HPV E6/E7 mRNA expression was also found, which also suggests a role for *TP53* R72 SNPs in the establishment of HPV-associated cancer disease, HNSCC included [55].

The prevalence of *TP53* R72 SNPs differs dependent on genetic descent, i.e., it is more prevalent in the northern hemisphere [24,33]. The distribution of these polymorphisms is likely bound to natural protection against ultraviolet radiation [24,33]. How this polymorphism affects HNSCC carcinogenesis is therefore not known, but explanations may range from a bystander effect to important clues of HNSCC generation. In any case, this might help explain why HNSCC cancer is more common among ethnic Caucasian individuals than ethnic Sub-Saharan Africans [24,56]. These hypotheses with respect to head and neck cancer and its possible association with *TP53* R72 SNPs, however, require further investigation.

We propose two distinct pathways where the *TP53* R72 SNPs increases the risk of HNSCC, both of which occur through dysregulation of the *TP53* pathway. It appears to increase the risk of persistent HPV infection, leading to an increased risk of HPV(+) OPSCC, and it is associated with increased *TP53* mutations, resulting in HPV(−) HNSCC (Figure 1).

Our methodology shows the robustness of studying somatic mutations in archival material, thus providing clinicians with a broader repertoire in clinical decision-making as well as forming a basis for research [17]. Regarding future investigations, it is possible to study whether *TP53* R72 or *TP53* somatic mutations are associated with a changed sensitivity to immune therapy. Thus, *TP53* status may become an even stronger part of precision medicine [43,57]. It is also of interest to study the role of *TP53* R72 SNPs in more detail in other SCCs [53,54,55]. Furthermore, whether an SCC lung tumor constitutes an HNSCC metastasis or a primary lung tumor is an important clinical question. In addition to HPV analysis, based on the large variation of *TP53* mutations found in our study, the determination of the exact *TP53* and other mutation(s) present in the tumor could help answer this.

## 5. Conclusions

In conclusion, we showed a higher prevalence of the codon 72 single-nucleotide polymorphism *TP53* R72 (i.e., proline → arginine) in HPV(+) OPSCC versus HPV(−) HNSCC and reported an association of the *TP53* R72 SNP with an increased prevalence of *TP53* somatic mutations in HPV(−) HNSCC. We have formulated our two main supported hypotheses in Figure 1. The *TP53* R72 SNP might contribute to *TP53* dysregulation through two pathways: either by increasing the risk of chronic HPV infection, leading to HPV(+) OPSCC, or by influencing the risk of acquiring TP53 mutations in HPV(−) HNSCC. The suggested mechanisms might shed some light on why Caucasians are more prone to HNSCC.

## Figures and Tables

**Figure 1 biomedicines-11-01838-f001:**
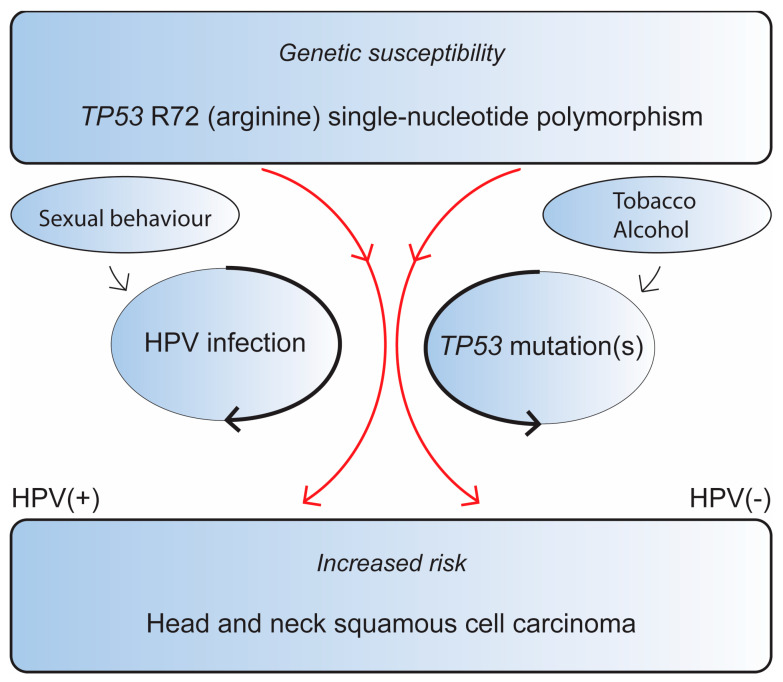
*TP53* R72 SNPs influence the risk of head and neck squamous cell carcinoma depending on the HPV status of the tumor. Red arrows indicate the SNPs dual pathway with regards to HPV-tumor status. Black arrows indicate that persistent HPV-infection and *TP53* mutations, respectively, further influence the risk of HNSCC development.

**Table 1 biomedicines-11-01838-t001:** Clinical patient characteristics by site/tumor HPV status upon diagnosis.

Variable		Oral	Other	Oropharynx HPV(−)	Oropharynx HPV(+)	Sign. HPV(−) vs. HPV(+) Condition
Age	Years Mean ± SD	60.6 ± 12.5	68.0 ± 3.2	65.1 ± 10.4	59.4 ± 10.1	*p* < 0.001
Gender	Males	28	4	27	37	n.s.
	Females	11	0	4	7	
T stage	1	9	2	1	5	n.s.
	2	13	1	12	21	
	3	2	0	6	6	
	4	5	1	6	11	
TNM n.a.				2	1	
N stage	0	21	2	11	9	*p* = 0.001
	1	2	0	2	4	
	2	6	1	9	27	
	3	0	1	3	3	
M stage	0	29	4	22	43	
	1	0	0	3	0	
Smoking	<10 tobacco years	15	0	2	17	n.s.
	>10 tobacco years	14	4	25	27	
Total number of patients		29	4	27	44	

Site according to International Classification Diseases, 10th edition; TNM stage according to the 7th TNM classification of malignant tumors of the International Union Against Cancer; n.a. = not applicable (three patients with metastasis without a primary tumor (C77.0) were not classified by TNM stage, but tumors were considered to originate from the oropharynx).

**Table 2 biomedicines-11-01838-t002:** Patients (*n*) with *TP53* mutation dependent on presence of *TP53* R72 SNP.

*TP53* R72 SNP (Arg)	*TP53* Mutations		Total Patients	Statistics by Mann–Whitney
	No	Yes		
All Included Patients				
No	15	8	23	
Yes	45	36	81	
Total patients	60	44	104	
Tumor HPV(−) patients				
No	11	7	18	Z = −3.41 *p* < 0.001
Yes	7	35	42	
Total patients	18	42	60	
Tumor HPV(+) patients				
No	4	1	5	
Yes	38	1	39	
Total patients	42	2	44	
Oropharynx tumor HPV(−) patients				
No	7	2	9	Z = −2.42 *p* = 0.016
Yes	5	13	18	
Total patients	12	15	27	
Oral cavity patients (all HPV(−))				
No	4	3	7	Z = −3.15 *p* = 0.002
Yes	1	21	22	
Total patients	5	24	29	

**Table 3 biomedicines-11-01838-t003:** Number of somatic *TP53* mutations in patients with *TP53* R72 SNP (Arg).

*TP53* R72 SNP	No. of *TP53* Mutations per Patient				Total Patients	Statistics by Mann–Whitney
	0	1	2	3		
All Patients Included						
No	15	6	2	0	23	
Yes	45	28	7	1	81	
Total patients	60	34	9	1	104	
Tumor HPV(−) all patients						
No	11	5	2	0	18	Z = −2.85 *p* = 0.004
Yes	7	28	6	1	42	
Total patients	18	33	8	1	60	
Tumor HPV(+) all patients						
No	4	1	0	0	5	
Yes	38	0	1	0	39	
Total patients	42	1	1	0	44	
Tumor HPV(−) oropharynx patients						
No	7	1	1	0	9	Z = −2.01 *p* = 0.044
Yes	5	12	1	0	18	
Total patients	12	13	2	0	27	
All oral cavity patients						
No	4	2	1	0	7	Z = −2.27 *p* = 0.023
Yes	1	15	5	1	22	
Total patients	5	17	6	1	29	

**Table 4 biomedicines-11-01838-t004:** *TP53* mutations in HPV(−) HNSCC patients (*n*) by tumor site (localization).

		Site			Total Patients	Mann–Whitney Statistics
		Oropharynx	Oral Cavity	Other		
*TP53* mutation	No	12	5	1	18	Z = −2.19 *p* = 0.028 (OP vs. OC)
	Yes	15	24	3	42	
Total		27	29	4	60	

**Table 5 biomedicines-11-01838-t005:** The presence of *TP53* R72 SNP according to tumor HPV status.

		Tumor HPV		Total Patients	Mann–Whitney Statistics
		No	Yes		
All patients					
*TP53* R72	No	18	5	23	Z = −2.25 *p* = 0.024
	Yes	42	39	81	
Total patients		60	44	104	
Oropharynx patients					
*TP53* R72	No	9	5	14	Z = −2.24 *p* = 0.025
	Yes	18	39	57	
Total patients		27	44	71	

**Table 6 biomedicines-11-01838-t006:** Clinical profile of *TP53* P72 (proline) HPV(+) tumors.

Patient Number	HPV Serotype	Any Pathogenic Mutation(s)	Age at Diagnosis	Smoking	Site	Site ICD 10
1	58	*TP53* TRAF-3	67	Yes	Tonsil	C09.9
2	16	FGFR-3	49	No	Base of tongue	C01
3	16	n.d.	84	Yes	Pharynx—overlapping	C10.8
4	16	n.d.	73	No	Tonsillar pillar	C09.1
5	16	n.d.	67	No	Tonsil	C09.9

n.d.= not detected.

## Data Availability

The data from this study are not allowed to be shared with anyone due to national legal regulations.

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
