# Peer review of "The TP53 Codon 72 Arginine Polymorphism Is Found with Increased TP53 Somatic Mutations in HPV(−) and in an Increased Percentage among HPV(+) Norwegian HNSCC Patients"

_biomedicines, 2023, doi:10.3390/biomedicines11071838_

Round 1

Reviewer 1 Report

The manuscript “The TP53 codon 72 arginine polymorphism is found with increased TP53 somatic mutations in HPV(-) and an increased percentage among HPV(+) Norwegian HNSCC patients” relates to identifying HPV tumor infection, the rate and repertoire of TP53 somatic mutations and the prevalence of the TP53 R72 (arginine) SNP, as well as the relation between these parameters in a cohort of both HPV(+) and HPV(-) HNSCC patients. Overall, this manuscript is a publishable work in Biomedicines with some value additions as mentioned below.

1. The full forms of the abbreviations used must be mentioned when mentioned for the first time. For example, HNSCC and its related terms.

2. Abstract: Please modify the abstract as follows for better clarity. a) An one-liner objective; b) Techniques used in methodology (1-2 lines); c) Conclusion in one or two sentences only.

3. The authors need to pay attention to the spacing between the words, typo errors, and grammar. This can be done by Grammarly software.

4. The authors need to discuss why Caucasians may be more prone to HNSCC concerning prior studies.

5. The utilization of the findings of this work concerning precision medicines will be appreciated.

6. A discussion about the prospects of this study and the benefits of this study will be useful to the readers.

7. The authors must highlight the technical advancement (inventiveness) of this work over previous studies.

Minor editing of English language required

Reviewer 2 Report

This topic about TP53 codon is very interesting, look at these points to improve it:

- Lines 98-104. It not clear in the aim of this paper how HPV tumor infection is related with head and neck cancer. This needs to be clarified at the end of the introduction section.

- Lines  213-220  "Somatic mutations in the TP53 gene are common in cancer [37,38]" In the discussion section it is important to discuss more about the role of TP53 in the development of tumors and the tumor microenvironment. Look at these very important papers: -- doi: 10.1016/j.clineuro.2021.106735  --  doi: 10.3389/fonc.2022.818693  --  doi: 10.3390/neurolint15020037  -- doi: 10.1186/s12974-023-02812-y

- Lines 236-238: "A limitation of our study is 236 that analysis was performed only in tumor cells. "  Explain more.

- Lines 254-260: "The prevalence of TP53 R72 differs depending on genetic descent; it is more prevalent 254 in the northern hemisphere [24,34]. This may co-explain why... " - what can the authors add new about this point? improve it

- Lines 272-274: "We have formulated our two main supported hy- 272 pothesis in the graphical abstract" Report some sentences here in the conclusion section.

Minor editing of English language required.

Round 2

Reviewer 2 Report

Authors solved all my criticisms.